# Putting DOAC Doubts to Bed(Side): Preliminary Evidence of Comparable Functional Outcomes in Anticoagulated and Non-Anticoagulated Stroke Patients Using Point-of-Care ClotPro^®^ Testing

**DOI:** 10.3390/jcm14155476

**Published:** 2025-08-04

**Authors:** Jessica Seetge, Balázs Cséke, Zsófia Nozomi Karádi, Edit Bosnyák, Eszter Johanna Jozifek, László Szapáry

**Affiliations:** 1Stroke Unit, Department of Neurology, University of Pécs, 7624 Pécs, Hungary; karadi.zsofia@pte.hu (Z.N.K.); bosnyak.edit@pte.hu (E.B.); jozifek.eszter@pte.hu (E.J.J.); 2Department of Emergency Medicine, University of Pécs, 7624 Pécs, Hungary; cseke.balazs@pte.hu

**Keywords:** acute ischemic stroke, direct oral anticoagulants, point-of-care testing, ClotPro^®^, functional outcome

## Abstract

**Background/Objectives:** Direct oral anticoagulants (DOACs) are now the guideline-recommended alternative to vitamin K antagonists (VKAs) for long-term anticoagulation in patients with non-valvular atrial fibrillation. However, accurately assessing their impact on ischemic stroke outcomes remains challenging, primarily due to uncertainty regarding anticoagulation status at the time of hospital admission. This preliminary study addresses this gap by using point-of-care testing (POCT) to confirm DOAC activity at bedside, allowing for a more accurate comparison of 90-day functional outcomes between anticoagulated and non-anticoagulated stroke patients. **Methods:** We conducted a retrospective cohort study of 786 ischemic stroke patients admitted to the University of Pécs between February 2023 and February 2025. Active DOAC therapy was confirmed using the ClotPro^®^ viscoelastic testing platform, with ecarin Clotting Time (ECT) employed for thrombin inhibitors and Russell’s Viper Venom (RVV) assays for factor Xa inhibitors. Patients were categorized as non-anticoagulated (*n* = 767) or DOAC-treated with confirmed activity (*n* = 19). Mahalanobis distance-based matching was applied to account for confounding variables including age, sex, pre-stroke modified Rankin Scale (mRS), and National Institutes of Health Stroke Scale (NIHSS) scores at admission and 72 h post-stroke. The primary outcome was the change in mRS from baseline to 90 days. Statistical analysis included ordinary least squares (OLS) regression and principal component analysis (PCA). **Results:** After matching, 90-day functional outcomes were comparable between groups (mean mRS-shift: 2.00 in DOAC-treated vs. 1.78 in non-anticoagulated; *p* = 0.745). OLS regression showed no significant association between DOAC status and recovery (*p* = 0.599). In contrast, NIHSS score at 72 h (*p* = 0.004) and age (*p* = 0.015) were significant predictors of outcome. PCA supported these findings, identifying stroke severity as the primary driver of outcome. **Conclusions:** This preliminary analysis suggests that ischemic stroke patients with confirmed active DOAC therapy at admission may achieve 90-day functional outcomes comparable to those of non-anticoagulated patients. The integration of bedside POCT enhances the reliability of anticoagulation assessment and underscores its clinical value for real-time management in acute stroke care. Larger prospective studies are needed to validate these findings and to further refine treatment strategies.

## 1. Introduction

### 1.1. The Challenge of Assessing DOAC Activity in Acute Stroke

Ischemic stroke remains a leading cause of morbidity worldwide [1], underscoring the importance of accurately identifying factors that influence patient recovery. Direct oral anticoagulants (DOACs), such as dabigatran, rivaroxaban, apixaban, and edoxaban, have fundamentally changed anticoagulation management [2,3] and are frequently recommended over traditional vitamin K antagonists (VKAs) in clinical guidelines [4], owing to their predictable pharmacokinetics, favorable safety profiles, and reduced need for routine laboratory monitoring [5,6,7,8].

Despite their widespread use, assessing the true impact of DOAC therapy on ischemic stroke outcomes remains methodologically challenging. The primary difficulty lies in accurately determining whether patients are actively anticoagulated upon hospital presentation [9]. Most studies classify patients as “on DOACs” based on self-reported medication use or prescription history; however, this approach does not reliably capture actual anticoagulant activity at the time of admission.

While international normalized ratio (INR) testing is well-established for monitoring VKAs, it is not a reliable marker of DOAC activity. Accurate assessment of DOAC levels requires specific assays: for dabigatran, dilute thrombin time (TT) [10] or ecarin-based assays [11] are considered most reliable, while calibrated anti-Xa assays are preferred for factor Xa inhibitors [12,13,14].

In the absence of these specialized tests, clinicians may rely on routine coagulation parameters as surrogate indicators. For dabigatran, TT or activated partial thromboplastin time (aPTT) are generally more informative than prothrombin time (PT)/INR [15,16], while for factor Xa inhibitors, PT/INR tends to be more reflective than aPTT [17,18]. However, these conventional assays have limited sensitivity to DOAC effects [19], and their clinical utility is further constrained by delays associated with centralized processing and limited availability outside regular working hours [20].

Consequently, the inability to rapidly and reliably confirm anticoagulation status has introduced misclassification bias in prior studies, contributing to inconsistent interpretations of the true efficacy and safety of DOAC therapy in acute ischemic stroke.

### 1.2. The Role of Point-of-Care-Testing

To address these challenges, rapid point-of-care testing (POCT) methods have gained increasing attention as a means of verifying anticoagulation status in acute clinical settings. Urine-based DOAC detection assays represent one such POCT approach and offer a quick and non-invasive alternative [21,22,23]. However, their diagnostic accuracy is limited by the inability to measure precise drug concentrations and by potential interference from dilution, cellular components (e.g., erythrocytes), and co-excreted substances such as bilirubin [24].

In recent years, viscoelastic POCT platforms such as rotational thromboelastometry (ROTEM^®^) [25] and thromboelastography (TEG^®^) [26], have been increasingly adopted for real-time coagulation assessment in acute care settings. ClotPro^®^ builds on these technologies with innovations that make it particularly suitable for evaluating anticoagulation status. While other systems such as ROTEM^®^ and TEG^®^ can be modified to detect DOAC effects, ClotPro^®^ offers standardized, drug-specific assays as part of its core test panel: ecarin Clotting Time (ECT) for direct thrombin inhibitors and Russell’s Viper Venom (RVV) for factor Xa inhibitors. These assays enable bedside confirmation of active anticoagulation within minutes, providing a practical tool for acute stroke management.

In this exploratory study, we assessed 90-day functional outcomes in ischemic stroke patients with objectively confirmed DOAC activity at admission, using the ClotPro^®^ POCT system. By comparing this subgroup to matched patients without anticoagulation, we aimed to generate preliminary insights into whether preexisting anticoagulation influences early stroke severity or functional recovery. Given the small sample size and the logistical constraints of real-time POCT in emergency care, this work is intended as a proof-of-concept to support future research on the role of DOACs and rapid diagnostics in acute stroke management.

## 2. Materials and Methods

### 2.1. Study Design and Setting

This retrospective cohort study was conducted at the Department of Neurology, University of Pécs. Data from ischemic stroke patients admitted between February 2023 and February 2025 were obtained from the Transzlációs Idegtudományi Nemzeti Laboratórium (TINL) STROKE-registry.

Inclusion criteria were (1) imaging-confirmed diagnosis of ischemic stroke via computed tomography or magnetic resonance imaging, (2) documented pre-stroke functional status using the modified Rankin Scale (mRS), (3) availability of National Institutes of Health Stroke Scale (NIHSS) scores at admission and 72 h post-stroke, and (4) complete 90-day mRS follow-up data.

### 2.2. Point-of-Care-Testing for Anticoagulation Verification

For patients with documented DOAC therapy, primarily prescribed for atrial fibrillation or other cardioembolic risk factors, anticoagulation status was assessed upon admission using the ClotPro^®^ viscoelastic testing platform (Haemonetics Corporation, Boston, MA, USA; formerly enicor GmbH, Munich, Germany). Two DOAC-specific assays were used: the ECT assay for detecting direct thrombin inhibitors and the RVV assay for identifying factor Xa inhibitors (apixaban, edoxaban, rivaroxaban). In accordance with the manufacturer’s recommendations, patients were classified as actively anticoagulated if the clotting time (CT) was ≥180 s on the ECT assay or ≥100 s on the RVV assay.

### 2.3. Outcome Measures

The primary outcome was the shift in mRS score from baseline (pre-stroke) to 90 days post-stroke, representing functional recovery or decline.

### 2.4. Statistical Analysis

Continuous variables were reported as mean ± standard deviation (SD) or median with interquartile range (IQR), depending on distribution. Categorical variables were summarized as frequencies and percentages. The Shapiro–Wilk test was used to assess normality. Between-group comparisons were made using Fisher’s exact test for categorical variables and either Student’s *t*-test or Mann–Whitney U test for continuous variables, as appropriate.

To ensure comparability between groups, Mahalanobis distance-based matching was performed, balancing the groups according to age, sex, pre-stroke mRS, admission NIHSS scores, and NIHSS scores at 72 h post-stroke. Post-matching balance was assessed using standardized mean differences (SMDs), with an SMD of <0.20 indicating acceptable balance.

Ordinary least squares (OLS) regression was used to assess the association between anticoagulation status and mRS-shift, adjusting for matched covariates. Principal component analysis (PCA) was performed to address multicollinearity and identify the most influential predictors of functional recovery. For both standard and PCA-derived regression models, 95% confidence intervals (CIs) were calculated to quantify the precision of predictor estimates. All analyses were conducted using Python version 3.13, with statistical significance set at *p* < 0.05.

### 2.5. Ethics Approval

The study protocol was reviewed and approved by the Scientific and Research Ethics Committee of the Medical Research Council of the University of Pécs (RRF-2.3.1-21-2022-00011, 1 September 2022) and the Scientific and Research Ethics Committee of the Medical Research Council of Hungary (BM/22444-1/2024, 1 September 2024). Informed consent was waived for this study as the data were collected as part of routine clinical documentation, in accordance with the institutional ethics approval.

## 3. Results

### 3.1. Demographic and Clinical Characteristics

Patients were categorized into two groups based on their anticoagulation status at admission: non-anticoagulated (*n* = 767) and DOAC-treated with confirmed active anticoagulation (*n* = 19) (Figure 1).

Prior to matching, significant baseline differences existed between DOAC-treated and non-anticoagulated patients (Table 1). Patients in the DOAC-treated group were significantly older (mean age 79.74 ± 9.40 years) compared to the non-anticoagulated group (69.40 ± 12.51, *p* < 0.001) and had slightly higher pre-stroke mRS scores (1.21 ± 1.27 vs. 0.61 ± 1.15, *p* = 0.057).

Additionally, cardioembolic stroke was significantly more prevalent in the DOAC group (63.2% vs. 26.9%, *p* < 0.001), and onset-to-door time was markedly shorter (105 [68–232] vs. 310 [102–832] minutes, *p* < 0.001).

Comorbidities such as hypertension and diabetes mellitus were also more frequent among DOAC-treated patients (hypertension: 100% vs. 79.8%, *p* = 0.020; diabetes: 57.9% vs. 33.6%, *p* = 0.047), whereas current smoking was more common in the non-anticoagulated group (31.9% vs. 0.0%; *p* = 0.002).

Regarding acute stroke treatment, DOAC-treated patients were significantly less likely to receive intravenous thrombolysis (IVT) (0.0% vs. 27.9%, *p* = 0.007), reflecting expected anticoagulation-related treatment limitations, while rates of mechanical thrombectomy (MT) were similar between groups (15.8% vs. 17.2%, *p* = 0.870).

After applying Mahalanobis distance-based matching, these baseline differences were effectively minimized, achieving balance across most covariates (Table 2, SMDs for all matched variables <0.20). The resulting groups were well-balanced in terms of age (79.72 ± 9.67 vs. 78.00 ± 8.17 years, *p* = 0.568) and pre-stroke mRS scores (1.28 ± 1.27 vs. 1.11 ± 1.23, *p* = 1.00).

However, some variables remained significantly different post-matching, including cardioembolic stroke prevalence (66.7% vs. 22.2%, *p* = 0.007), onset-to-door times (110 [72–221] vs. 408 [130–584] minutes, *p* < 0.001), and diabetes mellitus (55.6% vs. 16.7%, *p* = 0.035).

Additionally, as expected, DOAC-treated patients continued to be significantly less likely to receive IVT (0.0% vs. 33.3%, *p* = 0.007).

Despite these remaining differences, none of these factors were found to significantly predict 90-day functional outcomes in adjusted analyses (see Section 3.3). This reinforces that while DOAC-treated patients exhibited distinct baseline characteristics, these differences did not translate into worse functional outcomes.

### 3.2. Functional Outcomes

At 90 days post-stroke, functional outcomes were comparable between DOAC-treated and non-anticoagulated patients. Before matching, there was no significant difference in mean mRS-shift between the two groups (DOAC-treated: 1.89 vs. non-anticoagulated: 1.75; *p* = 0.771). Following matching, this similarity persisted, with the DOAC group showing a mean mRS-shift of 2.00 compared to 1.78 in the non-anticoagulated group (*p* = 0.745; Figure 2).

### 3.3. Ordinary Least Squares Regression Analysis and Functional Outcome Prediction

Regression analysis showed no significant association between confirmed active DOAC anticoagulation status and 90-day functional recovery, as measured by mRS-shift (*p* = 0.599). In contrast, NIHSS at 72 h (*p* = 0.004) and age (*p* = 0.015) emerged as significant independent predictors of functional outcome (Table 3). The model yielded an adjusted R^2^ of 0.541, indicating that a substantial proportion of the variability in recovery was explained, despite the inherent clinical complexity of the dataset.

### 3.4. Principal Component Analysis and Functional Outcome Prediction

To address multicollinearity among baseline variables and identify the most influential predictors of mRS-shift, PCA was conducted. This dimensionality reduction technique transformed the original set of correlated variables into orthogonal principal components (PCs), each capturing a distinct portion of variance within the dataset. The first eight PCs collectively explained 89.9% of the total variance, indicating that the majority of the original data structure was preserved (Table 4). Among these, PC1 and PC2 accounted for the greatest proportion of variance, 18.27% and 16.64%, respectively, suggesting they reflect the most prominent underlying dimensions associated with functional recovery.

### 3.5. Regression Analysis Using Principal Component Analysis-Derived Components

Regression analysis incorporating the PCA-derived components further confirmed that DOAC anticoagulation status was not significantly associated with 90-day functional recovery (*p* = 0.703). In contrast, PC2 emerged as the strongest independent predictor of mRS shift (*p* < 0.001), suggesting that this component captured the most relevant variance associated with post-stroke functional outcomes (Table 5).

Analysis of the PCA loadings (Figure 3) revealed that PC2 was primarily influenced by NIHSS at admission (loading: 0.62), NIHSS at 72 h (0.53), and cardioembolic stroke etiology (0.32). This loading pattern underscores the central role of stroke severity and early neurological status in determining functional recovery. The contribution of cardioembolic stroke to PC2 likely reflects its association with more severe clinical presentations, which are well-documented predictors of worse outcomes.

## 4. Discussion

### 4.1. Interpretation of Findings

Accurate determination of anticoagulation status in acute stroke patients remains a clinical challenge. Traditional laboratory assays, although analytically precise, often face practical limitations in emergency settings due to processing delays and limited availability outside routine hours. To address this, our study implemented POCT for bedside verification of DOAC activity, enabling immediate and objective classification of anticoagulation status at admission.

Despite initial baseline differences, such as older age, higher prevalence of cardioembolic stroke, and shorter onset-to-door times in the DOAC group, functional outcomes at 90 days were comparable between DOAC-treated and non-anticoagulated patients both before and after matching. Contrary to some earlier reports suggesting poorer prognosis in anticoagulated patients, our data demonstrated comparable recovery between groups. While anticoagulated patients exhibited baseline characteristics traditionally linked to worse outcomes, these differences did not translate into significantly worse functional recovery in our cohort. Importantly, after adjusting for major predictors such as stroke severity and age, anticoagulation status itself was not significantly associated with poorer recovery. These findings suggest that anticoagulation does not inherently impair post-stroke outcomes and emphasize that favorable recovery is achievable in anticoagulated patients when other clinical factors are well managed.

However, it is important to recognize that our assessment of anticoagulation was qualitative, based on functional coagulation assays (POCT), rather than precise quantitative measurement of plasma DOAC concentrations or adherence. Therefore, while evidence of anticoagulant effect at admission was confirmed, variability in dosing or adherence may exist and could influence outcomes. This limitation should be taken into account when interpreting our results, underscoring the need for further studies with detailed anticoagulant monitoring.

### 4.2. Mechanisms Underlying Observed Differences

To understand why outcomes were similar across groups despite baseline differences, we investigated the underlying factors influencing functional recovery.

Our analyses indicate that post-stroke functional outcomes are primarily influenced by stroke severity rather than anticoagulation status. This conclusion is supported by both OLS regression and PCA analyses. Specifically, PCA revealed that the second principal component was predominantly shaped by NIHSS scores at admission and 72 h, as well as cardioembolic stroke etiology. This suggests that although cardioembolic stroke contributes to outcome variability, its impact is largely mediated through the severity of neurological deficits characteristic of this subtype.

These findings align with prior research indicating that cardioembolic strokes are typically associated with greater initial neurological deficits [27] and larger infarct volumes [28,29], both of which are key predictors of poorer recovery, even when timely and appropriate treatment is administered.

### 4.3. Comparisons with Previous Research

Our findings align with and extend previous studies examining DOAC therapy effects and anticoagulation assessment in acute ischemic stroke. Seiffge et al. [30] and Küpper et al. [31] reported no significant differences in functional independence (mRS ≤ 2) between DOAC-treated and non-anticoagulated patients after adjustment for confounders. However, both studies primarily relied on self-reported medication use or electronic medical records to determine anticoagulation status. Although laboratory assays were occasionally used, testing was neither systematic nor consistently performed at admission, often with considerable delays that limited DOAC classification reliability.

Our study addresses this gap by employing POCT to confirm active anticoagulation status at the bedside. The ClotPro^®^ platform used has demonstrated strong concordance with conventional laboratory-based DOAC assays [32], and comparable performance to other POCT systems, as shown in studies by Yoshii et al. and Infanger et al. [33,34].

Moreover, ClotPro^®^ has proven clinically reliable in acute care, with validation in trauma settings [35], and ongoing evaluation in acute stroke care via the POINT STROKE trial [36]. Interim results from this prospective validation study, presented by Sedghi et al. at the European Stroke Organisation Conference (ESOC) 2025, demonstrated excellent diagnostic accuracy of the RVV assay for detecting factor Xa inhibitor activity in acute ischemic stroke patients. At a CT threshold of 110 s, the assay achieved an AUC of 0.97, correctly identifying active anticoagulation in 95.6% of cases, with sensitivity and specificity of 85.2% and 97.4%, respectively [37].

In addition to its strong diagnostic performance, POCT offers several qualitative advantages that are particularly valuable in the context of acute stroke care. ClotPro^®^ enables rapid bedside assessment of anticoagulation status, significantly reducing the time from patient arrival to clinical decision-making compared to conventional laboratory assays. This real-time availability of results supports urgent treatment decisions, including the safe administration of IVT or consideration of reversal agents in anticoagulated patients. Furthermore, the ClotPro^®^ system is user-friendly and well suited for use in emergency settings, requiring minimal training and producing interpretable results within minutes.

Taken together, the strong diagnostic performance and practical operational advantages of ClotPro^®^ underscore its value in acute stroke care, supporting integration into hyperacute workflows for efficient, consistent, and timely anticoagulation assessment and therapeutic decision-making.

### 4.4. Limitations and Future Directions

While our study offers valuable insights, several limitations warrant discussion and guide future research.

First, we did not measure DOAC plasma concentrations using liquid chromatography–tandem mass spectrometry, the gold standard for precise anticoagulant quantification. Although ClotPro^®^ provides practical and rapid bedside insight into anticoagulant effect, it reflects functional coagulation activity rather than directly quantifying plasma drug levels. Therefore, our assessment of “anticoagulated status” is based on coagulation function consistent with DOAC use rather than definitive laboratory confirmation.

Second, the relatively small size of the DOAC-treated cohort (*n* = 19) reflects real-world challenges in identifying and confirming active anticoagulation at stroke presentation. Among patients presenting with acute ischemic stroke, only a subset report DOAC use, and an even smaller fraction undergo POCT for anticoagulant assessment, as its use is still not routine and remains restricted by logistical constraints (e.g., time pressure, staffing, equipment availability). Even within the tested subgroup, some patients ultimately show no laboratory evidence of active anticoagulation despite self-reported DOAC intake, due to factors such as missed doses, non-adherence, or timing of last intake. Consequently, our final cohort includes only those patients with confirmed anticoagulant activity at admission, representing a particularly relevant group in which POCT has immediate clinical implications.

This multi-step selection process explains the limited sample size, which, while restricting statistical power, underscores the real-world complexity of anticoagulation assessment and the importance of objective testing. To reduce confounding, we applied Mahalanobis distance-based matching; however, the retrospective design still carries a risk of residual bias from unmeasured variables.

Third, our outcome analysis focused exclusively on the mRS-shift, a widely accepted but relatively coarse measure of functional recovery. Broader dimensions of post-stroke recovery, such as cognitive function, quality of life, psychological well-being, and long-term mortality, were not included in our analysis, which limits the comprehensiveness of our outcome assessment.

Future research should focus on prospective, multicenter studies with larger DOAC-treated cohorts to validate these findings and enable more detailed subgroup analyses based on stroke etiology and comorbidities. Expanding outcome measures beyond the mRS-shift to include patient-centered metrics such as quality of life, cognitive function, and survival would offer deeper insights into recovery. Additionally, comparative evaluations of different POCT platforms will be essential for standardizing anticoagulation assessment and optimizing their integration into acute stroke care pathways.

## 5. Conclusions

This preliminary analysis suggests that ischemic stroke patients with confirmed DOAC anticoagulation at admission may have 90-day outcomes comparable to non-anticoagulated patients. In our matched cohort, functional recovery appeared to be influenced primarily by stroke severity rather than anticoagulation status. Although limited by sample size, these findings demonstrate the feasibility of using viscoelastic POCT to objectively assess anticoagulation status in acute stroke settings and highlight the need for further research to refine management strategies for anticoagulated patients. Larger, prospective studies are needed to confirm these observations and explore the broader role of POCT in stroke care.

## Figures and Tables

**Figure 1 jcm-14-05476-f001:**
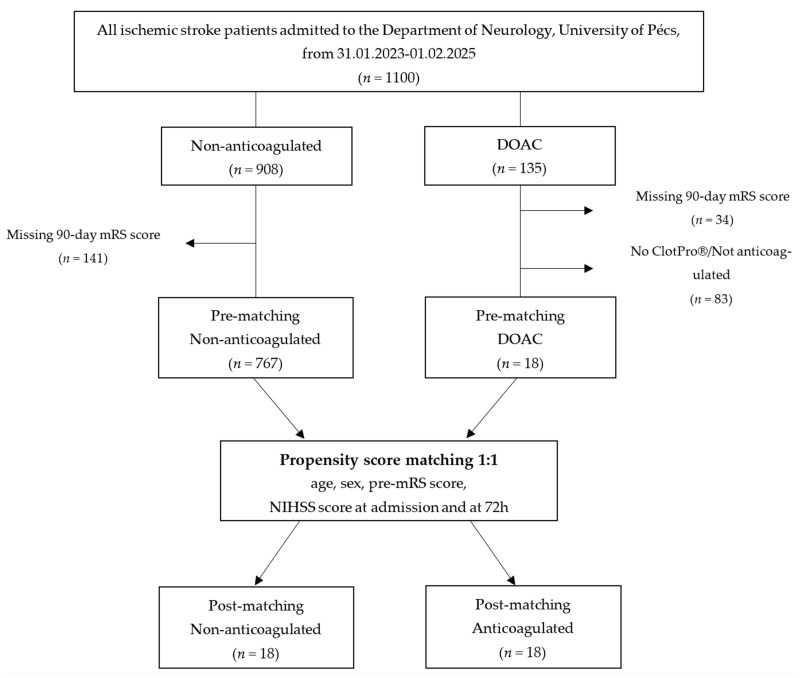
Flowchart of study. Abbreviations: DOAC = direct oral anticoagulant, mRS = modified Rankin Scale, pre-mRS = pre-morbid modified Rankin Scale, NIHSS = National Institutes of Health Stroke Scale at admission; note: “No ClotPro^®^/Not anticoagulated” refers to patients who either did not undergo ClotPro^®^ testing or whose ClotPro^®^ results showed no evidence of anticoagulation.

**Figure 2 jcm-14-05476-f002:**
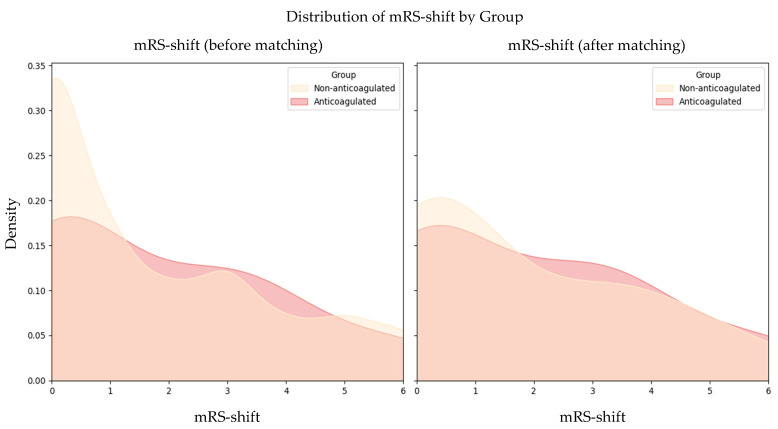
Distribution of mRS-shift by group (before and after matching). Abbreviations: mRS = modified Rankin Scale.

**Figure 3 jcm-14-05476-f003:**
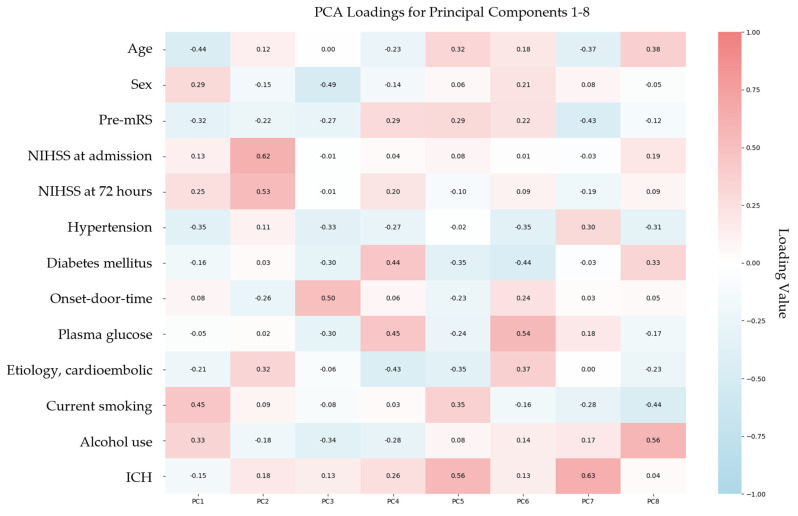
Heatmap of PCA loadings for the top eight components. Abbreviations: PCA = principal component analysis, PC = principal component, mRS = modified Rankin Scale, NIHSS = National Institutes of Health Stroke Scale, ICH = intracranial hemorrhage. Note: Each cell in the heatmap represents the strength and direction of correlation between a baseline variable and a principal component. Darker shades indicate stronger contributions, with positive and negative associations revealing how different variables group together within each component.

**Table 1 jcm-14-05476-t001:** Demographic and clinical characteristics before and after matching.

	Non-Anticoagulated (*n* = 767)	DOAC (*n* = 19)	*p*-Value	Matched Non-Anticoagulated (*n* = 18)	Matched DOAC (*n* = 18)	*p*-Value
Demographics						
Age (years), mean ± SD	69.40 ± 12.51	79.74 ± 9.40	<0.001 *	78.00 ± 8.17	79.72 ± 9.67	0.568
Sex, male, *n* (%)	378.0 (49.3%)	11.0 (57.9%)	0.494	10 (55.6%)	11 (61.1%)	1.000
Clinical Characteristics						
Pre-stroke mRS score, mean ± SD	0.61 ± 1.15	1.21 ± 1.27	0.057	1.11 ± 1.23	1.28 ± 1.27	0.692
NIHSS score at admission, mean ± SD	6.83 ± 6.17	7.11 ± 6.24	0.851	7.06 ± 6.31	7.28 ± 6.37	0.917
NIHSS score at 72 h, mean ± SD	5.44 ± 8.17	5.37 ± 9.77	0.975	5.06 ± 9.67	5.61 ± 9.99	0.866
ICH, *n* (%)	37 (4.8%)	0 (0.0%)	0.326	1 (5.6%)	0 (0.0%)	0.310
Etiology, cardioembolic, *n* (%)	206 (26.9%)	12 (63.2%)	<0.001 *	4 (22.2%)	12 (66.7%)	0.007 *
Onset-to-door time (min), median [IQR]	310 (102–832)	105 (68–232)	<0.001 *	408 (130–584)	110 (72–221)	<0.001 *
Plasma glucose (mmol/L), mean ± SD	7.74 ± 2.99	7.14 ± 2.21	0.381	7.41 ± 162	7.23 ± 2.23	0.993
Medical history, *n* (%)						
Hypertension	612 (79.8%)	19 (100%)	0.020 *	14 (77.8%)	18 (100.0%)	0.104
Diabetes mellitus	258 (33.6%)	11 (57.9%)	0.047 *	3 (16.7%)	10 (55.6%)	0.035 *
Current smoking	245 (31.9%)	0 (0.0%)	0.002 *	3 (16.7%)	0 (0.0%)	0.229
Alcohol use	332 (43.3%)	9 (47.4%)	0.816	13 (72.2%)	9 (50.0%)	0.305
Recanalization therapy, *n* (%)						
IVT	214 (27.9%)	0 (0.0%)	0.007 *	6 (33.3%)	0 (0.0%)	0.007 *
MT	132 (17.2%)	3 (15.8%)	0.870	3 (16.7%)	3 (16.7%)	1.000
IVT + MT	65 (8.5%)	0 (0.0%)	0.186	0 (0.0%)	0 (0.0%)	1.000

Abbreviations: DOAC = direct oral anticoagulant, SD = standard deviation, mRS = modified Rankin Scale, NIHSS = National Institutes of Health Stroke Scale, ICH = intracranial hemorrhage, IQR = interquartile range, IVT = intravenous thrombolysis, MT = mechanical thrombectomy; note: * indicates statistical significance at the 0.05 level.

**Table 2 jcm-14-05476-t002:** Standardized Mean Differences (SMDs) Before and After Matching.

	SMD Before Matching	SMD After Matching
Age	0.935	0.192
Sex	0.171	0.110
Pre-stroke mRS score	0.493	0.133
NIHSS score at admission	0.044	0.035
NIHSS score at 72 h	0.008	0.056

Abbreviations: SMD = standardized mean difference, mRS = modified Rankin Scale, NIHSS = National Institutes of Health Stroke Scale.

**Table 3 jcm-14-05476-t003:** Regression coefficients for predictors of mRS-shift.

Predictor	Coefficient	*p*-Value	95% CI
Anticoagulation status	0.4905	0.599	−1.433 to 2.414
Age	0.1188	0.015 *	0.026 to 0.211
Sex	−0.4080	0.629	−2.154 to 1.338
Pre-stroke mRS score	−0.1236	0.707	−0.804 to 0.557
NIHSS score at admission	−0.0083	0.942	−0.245 to 0.229
NIHSS score at 72 h	0.1930	0.004 *	0.070 to 0.316
ICH	−1.7137	0.439	−6.267 to 2.839
Etiology, cardioembolic	−1.1696	0.289	−3.419 to 1.080
Onset-to-door time	0.0004	0.583	−0.001 to 0.002
Plasma glucose	0.0944	0.682	−0.381 to 0.570
Hypertension	0.6411	0.624	−2.059 to 3.341
Diabetes mellitus	−0.7698	0.423	−2.740 to 1.201
Current smoking	−0.6960	0.599	−3.427 to 2.035
Alcohol use	−0.0914	0.915	−1.872 to 1.689
IVT	0.3097	0.826	−2.601 to 3.220
MT	0.6762	0.449	−1.161 to 2.513

Abbreviations: mRS = modified Rankin Scale, CI = confidence interval, NIHSS = National Institutes of Health Stroke Scale, ICH = intracranial hemorrhage, IVT = intravenous thrombolysis, MT = mechanical thrombectomy; note: * indicates statistical significance at the 0.05 level.

**Table 4 jcm-14-05476-t004:** Explained variance and cumulative variance for principal components.

Principal Component	Explained Variance (%)	Cumulative Variance (%)
PC1	18.27	18.27
PC2	16.64	34.90
PC3	14.35	49.26
PC4	11.52	60.78
PC5	9.89	70.67
PC6	8.25	78.93
PC7	5.98	84.90
PC8	5.02	89.92

Abbreviations: PC = principal component.

**Table 5 jcm-14-05476-t005:** Regression coefficients for principal components and predictors of mRS-shift.

Predictor	Coefficient	*p*-Value	95% CI
Anticoagulation status	−0.3264	0.703	−2.076 to 1.423
PC1	−0.0392	0.841	−0.439 to 0.360
PC2	0.9276	<0.001 *	0.495 to 1.360
PC3	0.1233	0.561	−0.309 to 0.556
PC4	0.2432	0.269	−0.201 to 0.688
PC5	0.0934	0.708	−0.415 to 0.602
PC6	0.1789	0.517	−0.384 to 0.742
PC7	−0.5224	0.106	−1.166 to 0.121
PC8	0.5545	0.100	−0.114 to 1.223
IVT	−1.1450	0.306	−3.407 to 1.117
MT	−0.4734	0.540	−2.046 to 1.099

Abbreviations: mRS = modified Rankin Scale, CI = confidence interval, PC = principal component, IVT = intravenous thrombolysis, MT = mechanical thrombectomy; note: * indicates statistical significance at the 0.05 level.

## Data Availability

The original contributions presented in the study are included in the article and further inquiries can be directed to the corresponding author.

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
