# Peer review of "Putting DOAC Doubts to Bed(Side): Preliminary Evidence of Comparable Functional Outcomes in Anticoagulated and Non-Anticoagulated Stroke Patients Using Point-of-Care ClotPro® Testing"

_jcm, 2025, doi:10.3390/jcm14155476_

Round 1
Reviewer 1 Report (New Reviewer)
Comments and Suggestions for Authors
This is an innovative review article that assesses the benefit of using point-of-care testing (POCT) to confirm DOAC activity at baseline/admission and to compare 90-day functional outcomes between anticoagulated and non-anticoagulated stroke patients in Hungary. The authors made a good effort in using tables to present the demographics and clinical characteristics of the study population. However, I recommend including a flowchart for better visualization of the study population, inclusion criteria/ selection process. I would recommend to explain why authors elected to use patients data only from 2023 to 2025.
The authors used good pictures/ images to demonstrate before-and-after mRS scores after the matching process. The discussion section is well organized, but I would suggest elaborating further on both the quantitative and qualitative aspects of point-of-care testing for better understanding about these tests.
Authors explained study limitations in a detailed manner.
Study conclusions are clear. Given the small sample size and the limited data range (February 2023 to February 2025), I agree that more prospective, large sample size, multicenter studies are needed to validate/support these findings. Overall this is a good review article.
Author Response
Please see the attachment.

Reviewer 2 Report (New Reviewer)
Comments and Suggestions for Authors
This is a well written and scientifically accurate manuscript. It adds to the literature in that it demonstrates that patients who have experienced a stroke who are not on DOAC have equal outcomes to those on DOAC in preliminary data. I think that this paper is good as is from a writing perspective and with some minor edits from the editorial team will fit well within the journal.
Author Response
Please see the attachment.

Reviewer 3 Report (New Reviewer)
Comments and Suggestions for Authors
This manuscript evaluates the impact of active DOAC therapy on stroke outcomes, using PCA and regression modeling to highlight the significance of stroke severity and cardioembolic etiology in functional recovery. The conclusion comes from a statistical analysis which has some robustness to be verified:
- In the manuscript, the ‘no independent association’ is misleading; people may think there is ‘dependent association’, better described as ‘no significant’.
- Can you explain why you selected those variables for matching as the confounding variables but did not include those with significant difference between non-anticoagulated and DOAC group, e.g, Diabetes mellitus?
- The sample size is too small, so I kind of doubt the results of regression with more than 15 predictors. Please do a diagnosis check for your regression model for the robustness: constancy of variance and normality check.
- The multicollinearity problem can be detected by VIF. You have selected the correlated variables subjectively. You may illustrate in detail why you selected those variables for the PCA, but kept others outside (IVT, MT).
I do recommend that you do a diagnosis check for your OLS model, and then use the VIF to select those important variables and use the rest of the variables to fit the OLS model again out of the interpretation, and of course, do a diagnosis again based on the small sample size.
Round 2
Reviewer 3 Report (New Reviewer)
Comments and Suggestions for Authors
In the revised manuscript, I see that some of the previous major concerns have been effectively addressed and improved. The author's responses have also resolved my main concerns. Apart from one minor detail that still needs to be revised, I have no further issues.
In the Discussion, it still shows “Importantly, after adjusting for major predictors such as stroke severity and age, anticoagulation status itself was not independently associated with poorer recovery.” Please change the “independently” here also.
Author Response
Please see the attachment.

This manuscript is a resubmission of an earlier submission. The following is a list of the peer review reports and author responses from that submission.
Round 1
Reviewer 1 Report
Comments and Suggestions for Authors
Dear authors,
thank you for conducting the present investigation on the preemptive effect of DOACs on severity and mid-term outcome of ischemic stroke.
Although interesting, there are some concerns:
-I would strongly recommend to remove the POCT (Clot-Pro) of the title, as the major part is distinguishing between anticoagulation and non-anticoagulation on admission only. There is no report on continuation of the anticoagulation neither was the device used to titrate these drugs.
-the guidelines you cite describe primary prevention in stroke patients. Aside, DOACs are recommended as secondary prevention after stroke. In that sense, I wonder, why a treatment for another reason should reduce the stroke severity.
-maybe report other important co-morbidities, too (kidney failure/disease, AF, vascular disease)
-please calculate a predictive score for thromboembolic events (e.g. Chads-Vascular score) to compare the groups regarding their risk (this comes out in your results, too).
-in the discussion, i am not fully convinced that the DOAC group did equally, as they had a higher risk profile. Please work this out in more detail.
-although the device might be a good one, avoid comparing it's function and use in other clinical scenarios as this is not matter of the study.
please find more comments in the attached pdf.

Although I not native speaker, there are certainly some areas of improvement.
An English language editing would be beneficial.
Reviewer 2 Report
Comments and Suggestions for Authors
Seetge et al. conducted a retrospective cohort study exploring the use of point-of-care ClotPro® testing to verify active direct oral anticoagulant (DOAC) therapy in ischemic stroke patients. While the study introduces important methodological innovations, particularly in implementing point-of-care testing (POCT), it faces several significant limitations that constrain its clinical impact and generalizability. The major Limitations:
Small Sample Size: Only 19 DOAC-treated patients were analyzed, compared to 767 non-anticoagulated controls. The authors acknowledge that this number reflects the real-world prevalence of this subpopulation, but it limits the statistical power and generalizability of their findings. Larger cohorts are needed to validate these results and perform more detailed subgroup analyses. The unique characteristics of the small DOAC group and retrospective design limit the applicability of the findings to broader stroke populations.
At present form, this results could be considered as preliminary and hypothesis-generating rather than definitive.
Larger, prospective multicenter studies are necessary to validate these findings and refine the role of POCT in stroke management.
Round 2
Reviewer 1 Report
Comments and Suggestions for Authors
Dear authors,
thanks for revising the manuscript.
Although you explained your position in the letter, I still feel there a careful revision and adoption of suggestions did not happen.
Comments on the Quality of English Language
some minor language edition would improve readability.
Reviewer 2 Report
Comments and Suggestions for Authors
The title is pretentious. It should be clearly stated in the title that the results are preliminary.